# Analytical Eddy Viscosity Model for Turbulent Wave Boundary Layers: Application to Suspended Sediment Concentrations over Wave Ripples

**Rafik Absi** [1,*] and **Hitoshi Tanaka** [2]

1   ECAM-EPMI—Ecole Supérieure d'Ingénieurs en Génie Electrique, Productique et Management Industriel, LR2E-Lab, Laboratory Quartz (EA 7393), 13 Boulevard de l'Hautil, 95092 Cergy-Pontoise, France
2   Institute of Liberal Arts and Sciences, Tohoku University, Sendai 980-8576, Japan
*   Correspondence: r.absi@ecam-epmi.com; Tel.: +33-669-155-620

**Abstract:** Turbulence related to flow oscillations near the seabed, in the wave bottom boundary layer (WBBL), is the phenomenon responsible for the suspension and transport of sediments. The vertical distribution of turbulent eddy viscosity within the WBBL is a key parameter that determines the vertical distribution of suspended sediments. For practical coastal engineering applications, the most used method to parameterize turbulence consists in specifying the shape of the one-dimensional-vertical (1DV) profile of eddy viscosity. Different empirical models have been proposed for the vertical variation of eddy viscosity in the WBBL. In this study, we consider the exponential-type profile, which was validated and calibrated by direct numerical simulation (DNS) and experimental data for turbulent channel and open-channel flows, respectively. This model is generalized to the WBBL, and the period-averaged eddy viscosity is calibrated by a two-equation baseline (BSL) k-ω model for different conditions. This model, together with a β-function (where β is the inverse of the turbulent Schmidt number), is used in modeling suspended sediment concentration (SSC) profiles over wave ripples, where field and laboratory measurements of SSC show two kinds of concentration profiles depending on grain particles size. Our study shows that the convection–diffusion equation, for SSC in WBBLs over sand ripples with an upward convection term, reverts to the classical advection–diffusion equation (ADE) with an "apparent" sediment diffusivity $\varepsilon_s^* = \alpha \, \varepsilon_s$ related to the sediment diffusivity $\varepsilon_s$ by an additional parameter $\alpha$ associated with the convective sediment entrainment process over sand ripples, which is defined by two equations. In the first, $\alpha$ depends on the relative importance of upward convection related to coherent vortex shedding and downward settling of sediments. When the convective transfer is very small, above low-steepness ripples, $\alpha \approx 1$. In the second, $\alpha$ depends on the relative importance of coherent vortex shedding and random turbulence. When random turbulence is more important than coherent vortex shedding, $\alpha \approx 1$, and "apparent" sediment diffusivity reverts to the classical sediment diffusivity $\varepsilon_s^* \approx \varepsilon_s$. Comparisons with experimental data show that the proposed method allows a good description of both SSC for fine and coarse sand and "apparent" sediment diffusivity $\varepsilon_s^*$.

**Keywords:** analytical model; eddy viscosity; turbulence; wave bottom boundary layer (WBBL); suspended sediments; concentrations; wave ripples; sediment diffusivity; vortex shedding; oscillatory flow

## 1. Introduction

Flows in the marine environment are complex with three-dimensional wave–current interactions, including irregular and breaking waves. The description of sediment transport in this environment becomes very complex. Near the seabed, flow oscillations are due to nearshore surface wave motions, hence the concept of the wave bottom boundary layer (WBBL) [1–9]. This layer plays an important role in the suspension and transport

of sediments above the seabed. Knowledge of the phenomena of erosion, entrainment, deposition, and resuspension of particles is linked to the description of the processes occurring in the WBBL. The oscillatory boundary layer associated with waves does not have enough time to develop before the flow is reversed. The thickness of this boundary layer remains in the order of a few centimeters, unlike the thickness of the current boundary layers, which are in the order of meters [5]. The bottom shear stress associated with waves generally dominates that associated with currents. The result is that the force acting on the sediments at the bottom is generally dominated by the wave. This phenomenon is explained by the fact that the oscillatory movement of the wave stirs up the bottom sediments, making them transportable by the currents. Very close to the bottom, the turbulence comes mainly from the oscillatory flow, and the predicted sediment concentrations are not different if the current is included or not [10].

The study of the WBBL is still the subject of much research [11–23]. Turbulence is the phenomenon responsible for the suspension of sediments. Different turbulence models have been used to solve the momentum and/or sediment concentration equations [24–26], and many possibilities have been proposed for parameterizing turbulence [5,27,28]. The simplest methods to parameterize the turbulence consist in specifying the shape of the one-dimensional-vertical profile of eddy viscosity, which is called "specific" or "specified" eddy viscosity models [29]. Different assumptions have been made about the vertical variation of eddy viscosity in the WBBL, and empirical models have been proposed (Figure 1) [1–3,30–32]. Another eddy viscosity profile is the well-known parabolic profile for open-channel flows [33] based on log-law velocities, which was used for turbulent wave boundary layers by replacing the water depth with the wave boundary layer thickness [5] (lower part of Myrhaug profile [3] in Figure 1). Another formulation given by the exponential-type profile, which was first proposed empirically for the planetary boundary layers, was applied to turbulent wave boundary layers and calibrated by $k$-$\varepsilon$ [34,35] and $k$-$\omega$ [36] turbulence closure schemes.

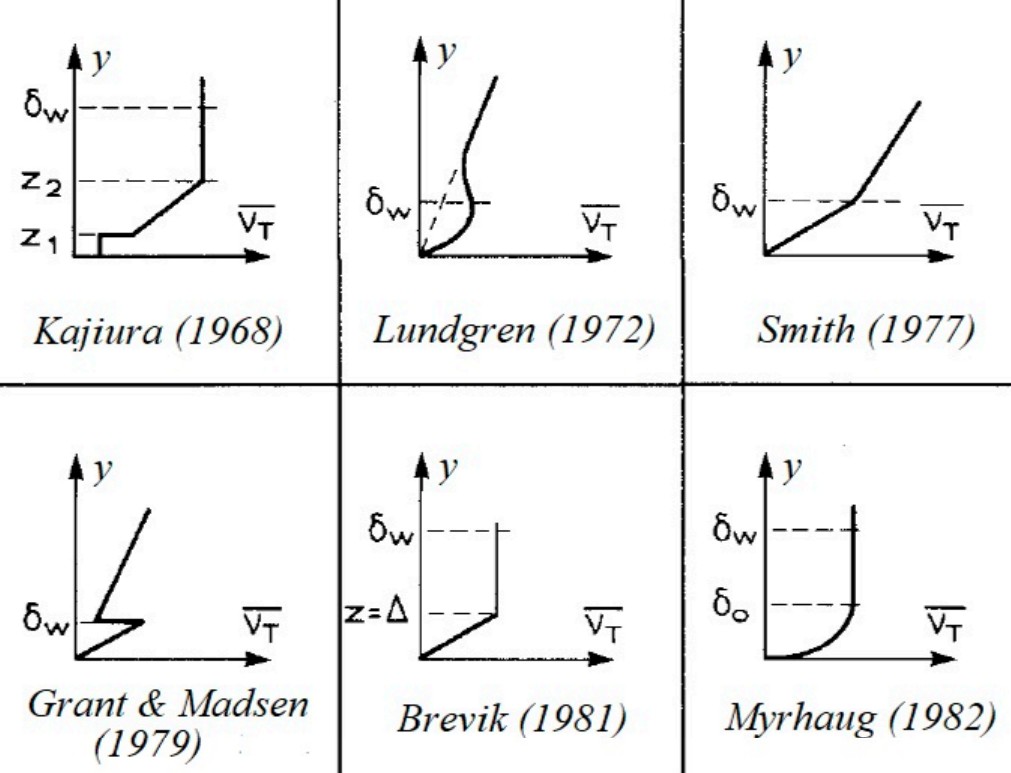

**Figure 1.** Assumptions for the vertical variation of eddy viscosity in the bottom boundary layer, (adapted with permission from [5]).

Six models for wave bottom boundary layer flows were assessed and compared to laboratory data [29]. The models were a laminar and five one-dimensional-vertical eddy viscosity models given, respectively, by two specific eddy viscosity models—linear and parabolic—and three turbulence closure models—$k$-one equation, $k$–$\varepsilon$, and $k$–$\omega$ two-equation turbulence closure models. The results show that, for velocity profiles, the linear model was more accurate, and the least accurate was the $k$–$\varepsilon$ models, while for bed shear stress and TKE, the $k$–$\omega$ model shows the best results [29]. However, other specific eddy viscosity models can provide a link to the k-equation and therefore to the turbulence closure schemes [37].

From the different specific eddy viscosity models for the WBBL, two profiles show particular interest in engineering applications: the parabolic-uniform profile [3,7,38,39] and the exponential-type or exponential–linear profile [34–36,40–42]. These two eddy viscosity profiles were analyzed and assessed [36] through (1) investigation of eddy viscosity in steady fully developed plane-channel flow, and (2) comparisons with numerical results of the two-equation baseline (BSL) $k$-$\omega$ model [11,43]. These studies show that these two profiles are able to describe the eddy viscosity distribution in the wave bottom boundary layer but for different wave conditions given by the roughness parameter $a_m/k_s$, where $a_m$ is the wave orbital amplitude and $k_s$ the equivalent roughness. The study concluded that the exponential-type profile is adequate for $a_m/k_s < 500$, while the parabolic-uniform profile is more suitable for $a_m/k_s \geq 500$ [36].

The vertical distribution of turbulent eddy viscosity within the WBBL generated by waves in shallow waters is a key parameter and the main quantity that determines the vertical distribution of suspended sediments [5–7]. In coastal zones, sediment transport modeling is important for the predictions of coastline evolution and seabed changes [44–48]. In the coastal engineering approach, the net total coastal sediment transport (averaged over the wave period) is obtained as the sum of the net bed load and net suspended load transport rates. For suspended load, the net sand transport is defined as the sum of the net current-related and the net wave-related transport components. The wave-related suspended sediment transport is defined as the transport of suspended sediments by the oscillatory flows. The prediction of wave-related suspended transport components is based on the widely and well-known gradient diffusion model. The resolution of the related classical one-dimensional vertical (1DV) advection–diffusion equation (ADE) needs sediment diffusivity $\varepsilon_s$ and settling velocity of sediments $\omega_s$. The diffusivity of sediments $\varepsilon_s$ is related to the diffusivity of momentum, i.e., the eddy viscosity $\nu_t$, by a coefficient $\beta = \varepsilon_s/\nu_t$ (i.e., the inverse of the turbulent Schmidt number).

Four models of sediment transport were compared to observations made in wave and current flows above plane beds [49]. The results show that the "STP" model, based on a vertical distribution of eddy viscosity given by a parabolic shape [27], seems to give interesting results similar to that of a more complex model, namely the $k$-$L$ type second-order closure model [50]. On the other hand, the results from the Reynolds stress model [51] based on a simplified two-phase flow approach do not improve the predictions of other simpler models [49].

In oscillatory flows, cycle-mean sediment diffusivity above ripples is significantly greater than the cycle-mean eddy viscosity, i.e., $\beta > 1$ [6,52–54]. The value of β was suggested empirically as a constant equal to four ($\beta = 4$) for rippled beds [6] and the near-bed sediment diffusivity a constant equal to $\varepsilon_s = 0.061 k_s U_0$. Another empirical sediment diffusivity formulation involves a three-layer distribution [38] (Figure 2).

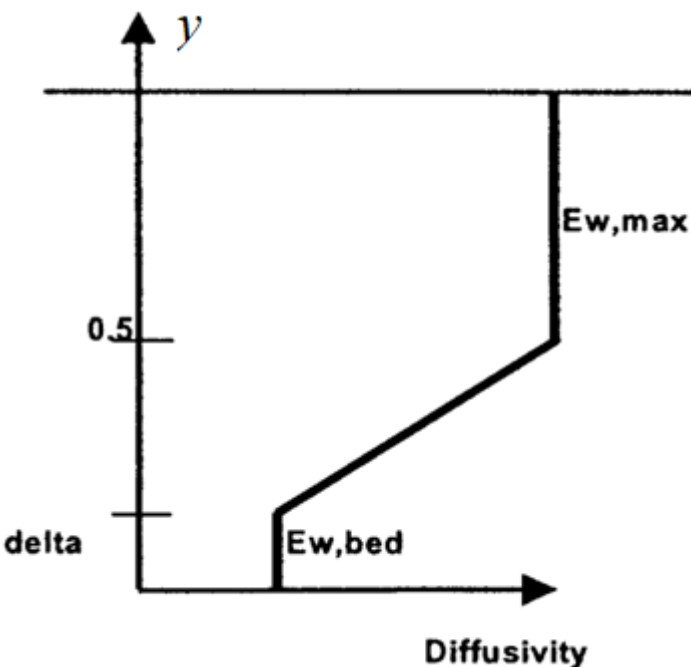

**Figure 2.** Vertical distribution of wave-related sediment diffusivity in the bottom boundary layer (adapted with permission from [38]).

For suspended sediments in oscillatory flows over sand ripples, field and laboratory data show, for time-averaged concentration profiles in semi-log plots, a contrast between upward convex profiles for fine sand and upward concave profiles for coarse sand [55–58]. Careful examination of experimental data for coarse sand shows a near-bed upward convex profile beneath the main upward concave profile [42,54].

The above review shows that:

- The different comparative studies show that even if more complex models contain more information about turbulence, they do not always provide the best results compared to simpler models. Therefore, complexity does not systematically imply superiority, in particular in coastal engineering practice, where simple models are often preferred for practical applications.
- Different assumptions were made about the vertical variation of eddy viscosity and sediment diffusivity in the WBBL, and empirical models were proposed. It is important to know which one is the best and to find the link with turbulence closure schemes.
- Even if the vertical distribution of sediment diffusivity (Figure 2) seems similar to the three-layer distribution of eddy viscosity of Kajiura (Figure 1), it is very different from the other eddy viscosity profiles, especially the two that show interest, namely, the parabolic-uniform and exponential-type profiles [36], taking into account their close link to results from turbulence closure schemes. In addition, the discontinuous three-layer distribution is mainly the result of an empirical approach, while theoretical models provide analytical continuous solutions without discontinuities as in the three-layer profile. It is important to find the link between eddy viscosity and sediment diffusivity profiles.

The aim of this study is to:

- Provide a unique explanation/interpretation for the different eddy viscosity and sediment diffusivity data for the WBBL.
- Replace former different empirical profiles of eddy viscosity/sediment diffusivity with a unique analytical/theoretical model. Following our former study [36], the one-dimensional-vertical profile of eddy viscosity will be investigated based on new results obtained from an analytical study of eddy viscosity in steady fully devel-

oped plane-channel and open-channel flows [59–61]. The selected analytical model, namely, the exponential-type profile, is first validated by direct numerical simulation (DNS) and experimental data of steady plane-channel and open-channel flows, respectively [60,61]. It will be generalized to the WBBL, assessed, and calibrated by comparisons with numerical results of the two-equation baseline (BSL) *k-ω* model. A new calibration of the period-averaged eddy viscosity for oscillatory flows for different wave conditions through the parameter $a_m/k_s$ will be proposed.

- Use the proposed analytical eddy viscosity model in the computation of suspended sediment concentration profiles in oscillatory flow over sand ripples.

Two models of suspended sediment concentrations are presented: the classical advection–diffusion equation based on the gradient diffusion model and the convection–diffusion equation for suspended sediments in oscillatory flows over sand ripples with an upward convection term. Both models need the sediment diffusivity, which is related to the proposed exponential-type profile of eddy viscosity and the turbulent Schmidt number. Profiles for suspended sediment concentration and sediment diffusivity will be presented.

## 2. Eddy Viscosity Formulation

Among the different eddy viscosity profiles, the well-known parabolic eddy viscosity profile, which is largely used for open-channel flows [33], was adapted to turbulent wave boundary layers [5] by replacing the water depth with the wave boundary layer thickness. Following this example, we will first show the interest of the exponential-type eddy viscosity profile in turbulent channel and open-channel flows, then we will generalize this analytical model to turbulent wave boundary layers.

### 2.1. Eddy Viscosity Formulation for Steady Channel and Open-Channel Flows

More theoretical analytical eddy viscosity models are based on the concepts of velocity and length scales [62–66], which are related to the exponentially decreasing turbulent kinetic energy (TKE) function, and mixing length, namely, the exponential-type profile of eddy viscosity [59–61], given by

$$\nu_t(y) = u_* y e^{-\frac{y^+ + 0.34\mathrm{Re}_* - 11.5}{0.46\mathrm{Re}_* - 5.98}} \, ,$$ (1)

where $y$ is the vertical distance from the bed/bottom and in wall units, $y^+ = y u_*/\nu$, $\mathrm{Re}_* = h u_*/\nu$ is the friction Reynolds number, $u_*$ is the friction or shear velocity, $\nu$ is the kinetic viscosity, and $h$ is the half channel height or the flow depth.

This $Re_*$-dependent eddy viscosity (Equation (1)) was validated through computation of velocity profiles and comparisons to direct numerical simulation (DNS) and experimental data of both velocities and eddy viscosity (Figure 3) [59–61]. It is possible to write Equation (1) in the following form

$$\nu_{t\,a}(\xi) = C_\alpha \, \xi e^{-C_1 \xi}$$ (2)

where $\xi = y/h$, $\nu_{t\,a} = \nu_t/(u_* h)$ and the two coefficients $C_\alpha$ and $C_1$ are given by $C_\alpha = e^{-\frac{0.34\mathrm{Re}_* - 11.5}{0.46\mathrm{Re}_* - 5.98}}$ and $C_1 = \frac{\mathrm{Re}_*}{0.46\mathrm{Re}_* - 5.98}$. For large values of $Re_*$ ($Re_* > 2000$), the model (Equation (1)) becomes $Re_*$-independent, and the two coefficients $C_\alpha = \alpha_1 \kappa$ and $C_1$ reach asymptotic values equal, respectively to $C_\alpha = 0.477$ and $C_1 = 2.17$ [61].

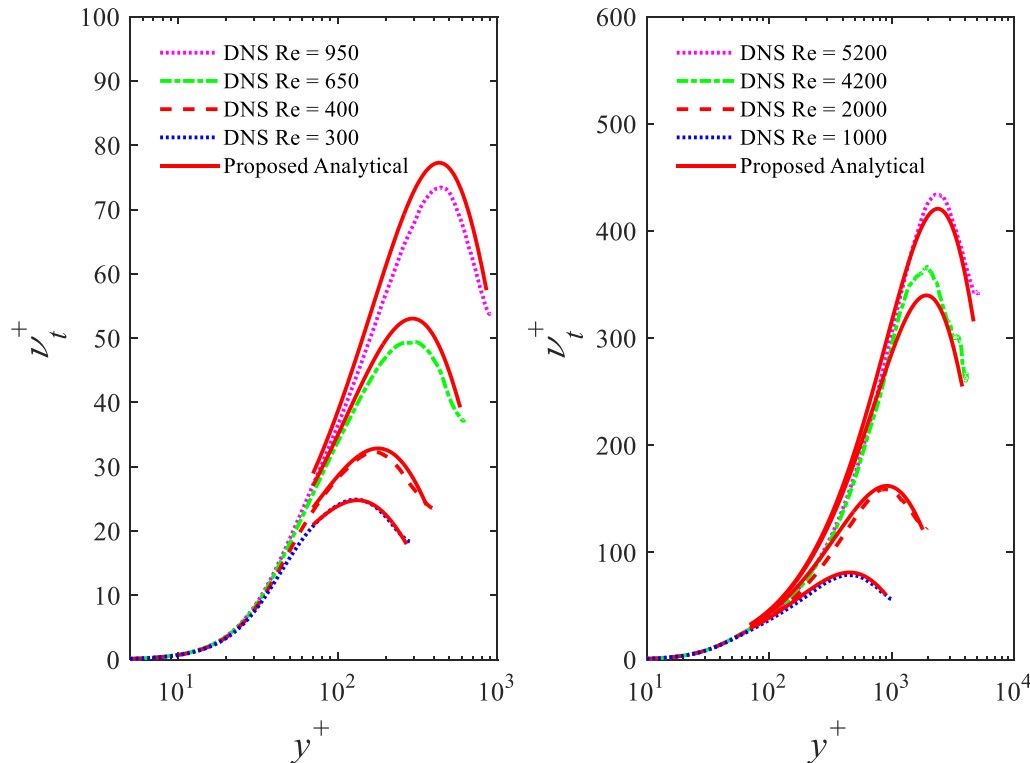

**Figure 3.** Eddy viscosity profiles in turbulent channel flows for different friction Reynolds numbers $\nu_t{}^+ = \nu_t/\nu$; red solid lines: Equation (1); dotted and dashed lines: DNS data [60].

With an additional correction to account for the damping effect of turbulence near the free surface, we used [61] a damping function in order to decrease turbulent viscosity near the free surface as

$$\nu_{t\,a}(\xi) = C_\alpha\,\xi e^{-C_1\xi}\left(1 - e^{-B_f(1-\xi)}\right) \tag{3}$$

Figure 4 shows the eddy viscosity profiles given by Equation (2) (red solid line) and Equation (3) (red dashed line) and comparisons with parabolic and wake-modified profiles. The profile of Equation (3) is similar to the wake-modified profile with a wake parameter $\Pi = 2$ (used for open-channel flows [33]). Equations (2) and (3) provide identical results for $\xi < 0.4$, and therefore Equation (2) could be used for sediment transport modeling.

### 2.2. Eddy Viscosity Formulation for Oscillatory Flows

In this study, we consider a sinusoidal oscillatory flow, where the free stream velocity is given by

$$U(t) = U_m\,\sin(\sigma\,t)$$

where $U_m$ is the maximum value of the free stream velocity (or the velocity at wave crest) and $\sigma = 2\pi/T$ is the angular frequency. The different flow conditions are given by the roughness parameter $a_m/k_s$, where $k_s$ is Nikuradse's equivalent sand roughness and $a_m = U_m/\sigma$ is the orbital amplitude of fluid just above the boundary layer or near-bed semi-excursion.

#### 2.2.1. Analytical Eddy Viscosity Model

Equation (2) of eddy viscosity was validated for steady plane-channel and open-channel flows. However, for use in wave boundary layers, we need to assess this equation for the case of oscillatory flows. For oscillatory flows, Equation (2) is written in the following dimensionless form

$$\nu_{t\,a}(\xi) = \frac{\nu_t}{U_m\,a_m} = C_\alpha\,\xi e^{-C_1\xi} \tag{4}$$

where $\xi = y/y_h$ and $y_h$ is the distance from the wall to the axis of symmetry for oscillatory water tunnel experiments.

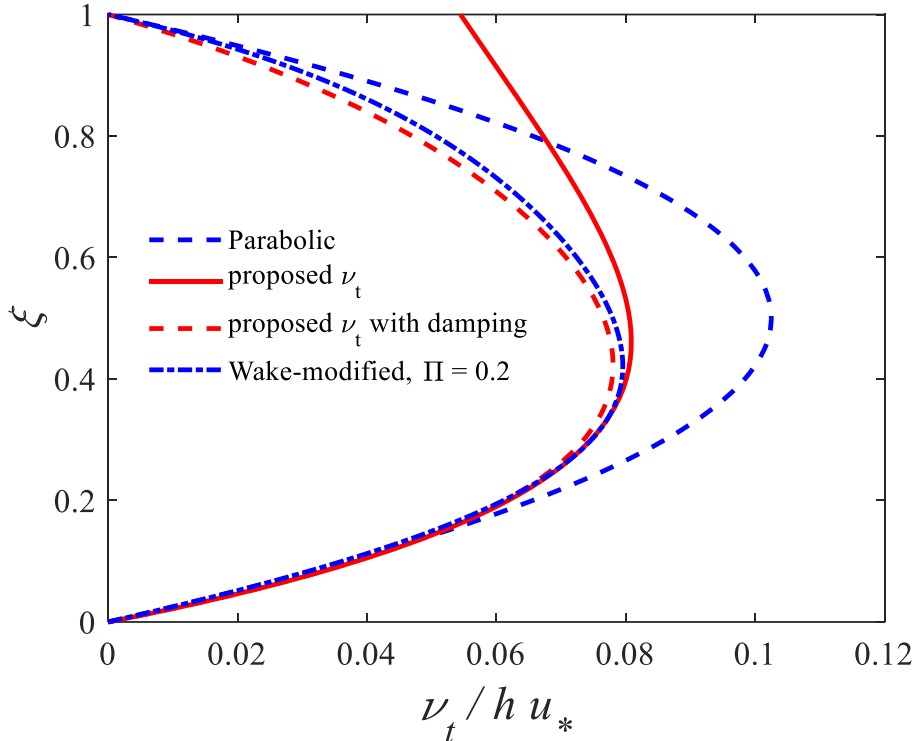

**Figure 4.** Eddy viscosity profiles in open-channel flows; red solid line: eddy viscosity (Equation (2)) with $C_\alpha = 0.477$ and $C_1 = 2.17$; red dashed line: eddy viscosity with free surface damping function (Equation (3)) with $B_f = 6$; blue dashed line: parabolic eddy viscosity; blue dash-dotted line: log-wake-modified with $\Pi = 0.2$ [61].

The profile given by Equation (4) was used for turbulent wave boundary layers [34–36,42,56,57] and calibrated by $k$-$\varepsilon$ [34,35] and $k$-$\omega$ [36] turbulence closure schemes. Equation (4) was also used indirectly in [56] for the validation of a finite-mixing-length theory, which needs vertical profiles of mixing velocity and mixing length. Theses profiles were given by mixing length increases linearly and mixing velocity decreases exponentially with $\xi$, and therefore the shape of Equation (4).

The value of the eddy viscosity should vanish outside the bottom boundary layer. This is possible by using the damping function (of Equation (3)) as for damping effect near the free surface in open-channel flows. Therefore, it is possible to write Equation (4) in the following form

$$\nu_{t\,a}(\xi) = \frac{\nu_t}{U_m\,a_m} = C_\alpha\,\xi e^{-C_1\xi}\left(1 - e^{-B_f(1-\xi)}\right) \tag{5}$$

However, since both Equations (4) and (5) provide identical results for $\xi < 0.4$ (as for open-channel flows, Figure 4), Equation (4) could be used for sediment transport modeling.

### 2.2.2. Baseline (BSL) k-ω Model

Equation (4) is analyzed by the baseline (BSL) k-ω model [43], which allows accurate prediction of velocity profiles in WBBLs [11]. The BSL $k$–$\omega$ model is a two-equation model that gives results similar to the $k$–$\omega$ model of [67] in the inner boundary layer (Equations (6)–(8)) but changes gradually to the $k$–$\varepsilon$ model [68] towards the outer boundary layer and the free stream velocity. The blending between the two regions is achieved by a blending function $F_1$, changing gradually from one to zero in the desired region.

Equations of turbulent kinetic energy $k$, specific dissipation $\omega$, and eddy viscosity are given, respectively, by:

$$\frac{\partial k}{\partial t} = \frac{\partial}{\partial y}\left((\nu + \nu_t\sigma_{k\omega})\frac{\partial k}{\partial y}\right) + \nu_t\left(\frac{\partial u}{\partial y}\right)^2 - \beta^*\omega k \tag{6}$$

$$\frac{\partial \omega}{\partial t} = \frac{\partial}{\partial y}\left((\nu + \nu_t\sigma_\omega)\frac{\partial \omega}{\partial y}\right) + \gamma\left(\frac{\partial u}{\partial y}\right)^2 - \beta\omega^2 + 2(1-F_1)\sigma_{\omega2}\frac{1}{\omega}\frac{\partial k}{\partial y}\frac{\partial \omega}{\partial y} \tag{7}$$

$$\nu_t = \frac{k}{\omega} \tag{8}$$

The constants of the model are given by $\sigma_{k\omega} = 0.5$, $\beta^* = 0.09$, $\sigma_\omega = 0.5$, $\gamma = 0.553$, $\beta = 0.075$ [11].

### 2.2.3. Boundary Conditions and Numerical Method

At the bed, the no-slip condition is assumed, thus the velocities and turbulent kinetic energy are $u = k = 0$, and at the axis of symmetry (outside boundary layer), the gradients of velocity, turbulent kinetic energy, and specific dissipation rate are equal to zero, i.e., at $y = h$, $\partial u/\partial y = \partial k/\partial y = \partial \omega/\partial y = 0$, and the velocity at the outside boundary layer is equal to the free stream velocity ($u = U$). The wall boundary condition of Wilcox (1988) is used to express the effect of roughness in the present model [11].

The nonlinear governing equations were solved using a Crank–Nicolson type implicit finite-difference scheme. The grid spacing increases exponentially in order to achieve better accuracy near the wall. In space 100 and in time 7200 steps per wave cycle were used. The convergence was achieved through two stages based, respectively, on the dimensionless values of $u$, $k$, and $\omega$ and on the maximum wall shear stress in a wave cycle. For both the stages, the convergence limit was set to $1 \times 10^{-6}$. The instantaneous value of $\nu_t$ can be obtained numerically from Equations (6)–(8) [11].

### 2.2.4. Results and Calibration of the Analytical Eddy Viscosity Model

Figure 5 presents the temporal and spatial variation of dimensionless eddy viscosity for a sinusoidal wave. Figure 6 shows the comparison between period-averaged eddy viscosity obtained from the BSL $k$-$\omega$ model (symbols) and analytical profile of Equation (4) (curves). Even if the eddy viscosity is highly time-dependent (Figure 5), the period-averaged dimensionless eddy viscosity (Figure 6) has a shape that is well described by the exponential-type analytical profile given by Equation (4) for two flow conditions given by the roughness parameter $a_m/k_s$ equal to 100 and 300, respectively. For $a_m/k_s = 100$, Equation (6) allows a good description of the period-averaged dimensionless eddy viscosity (Figure 6) for $\xi < 0.6$. However, for $a_m/k_s = 300$, data are well described over the entire height.

In our former study [36], the two parameters were calibrated by the BSL k-$\omega$ model up to $a_m/k_s = 500$. The values were

$$\alpha = 0.2(a_m/k_s)^{-0.97} \tag{9}$$

with $C_\alpha = \alpha\kappa$

$$C_\alpha = 0.082\,(a_m/k_s)^{-0.97} \tag{10}$$

and

$$C_1 = 29.7(a_m/k_s)^{-0.52} \tag{11}$$

In order to allow more accurate equations, a second calibration is proposed based on comparisons of Equation (4) with data from BSL k-$\omega$ model for more flow conditions up to $a_m/k_s = 5000$ (Figure 7).

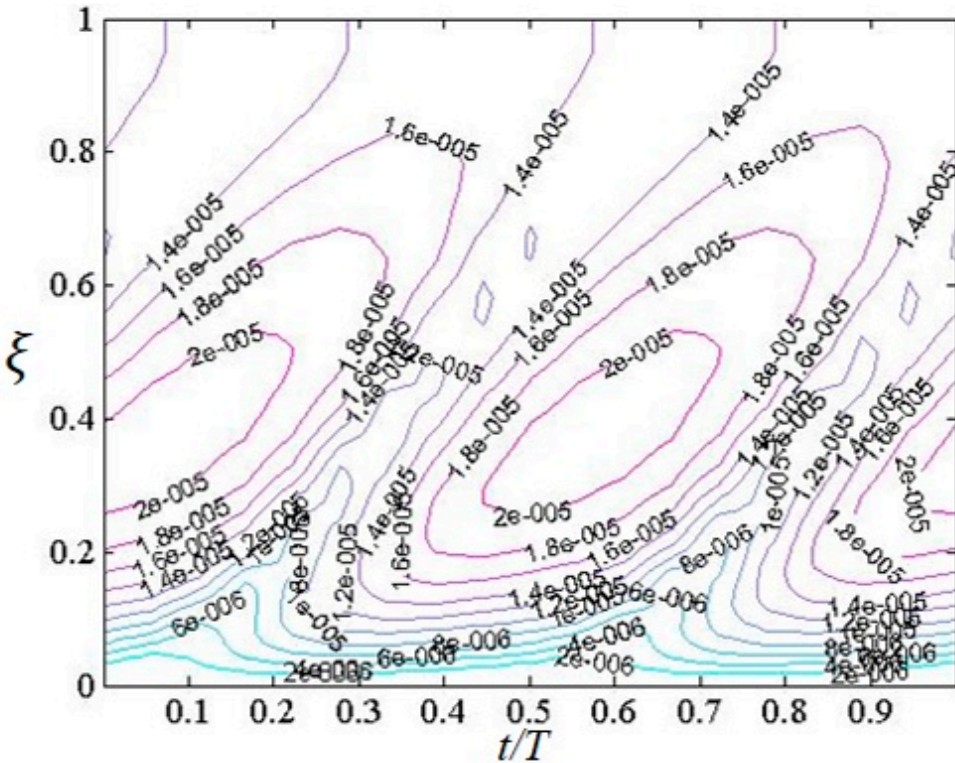

**Figure 5.** Temporal and spatial variation of dimensionless eddy viscosity for a sinusoidal wave obtained by the BSL k-ω model. Flow conditions: $U_m = U_0 = 3.63$ m/s; $a_m = 1.73$ m; T = 3 s; $k_s = 1.5$ cm; Re = 437000.

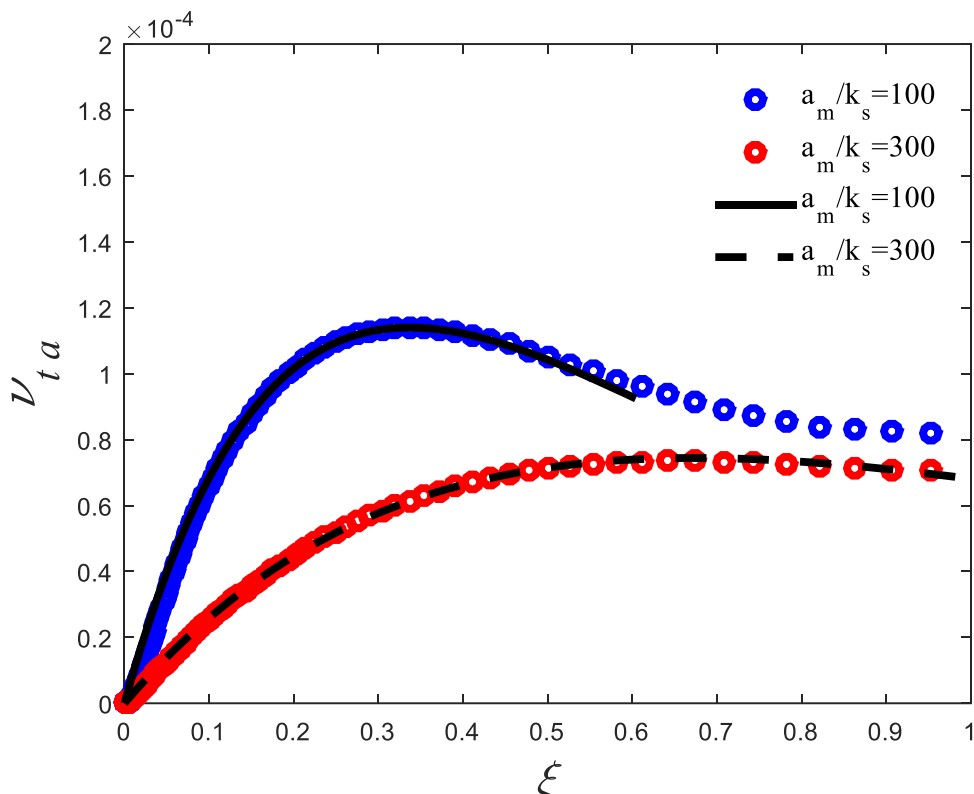

**Figure 6.** Period-averaged dimensionless eddy viscosity for a sinusoidal wave, curves: Equation (4), symbols: data from the BSL k-ω model, $a_m/k_s = 100$ and $a_m/k_s = 300$.

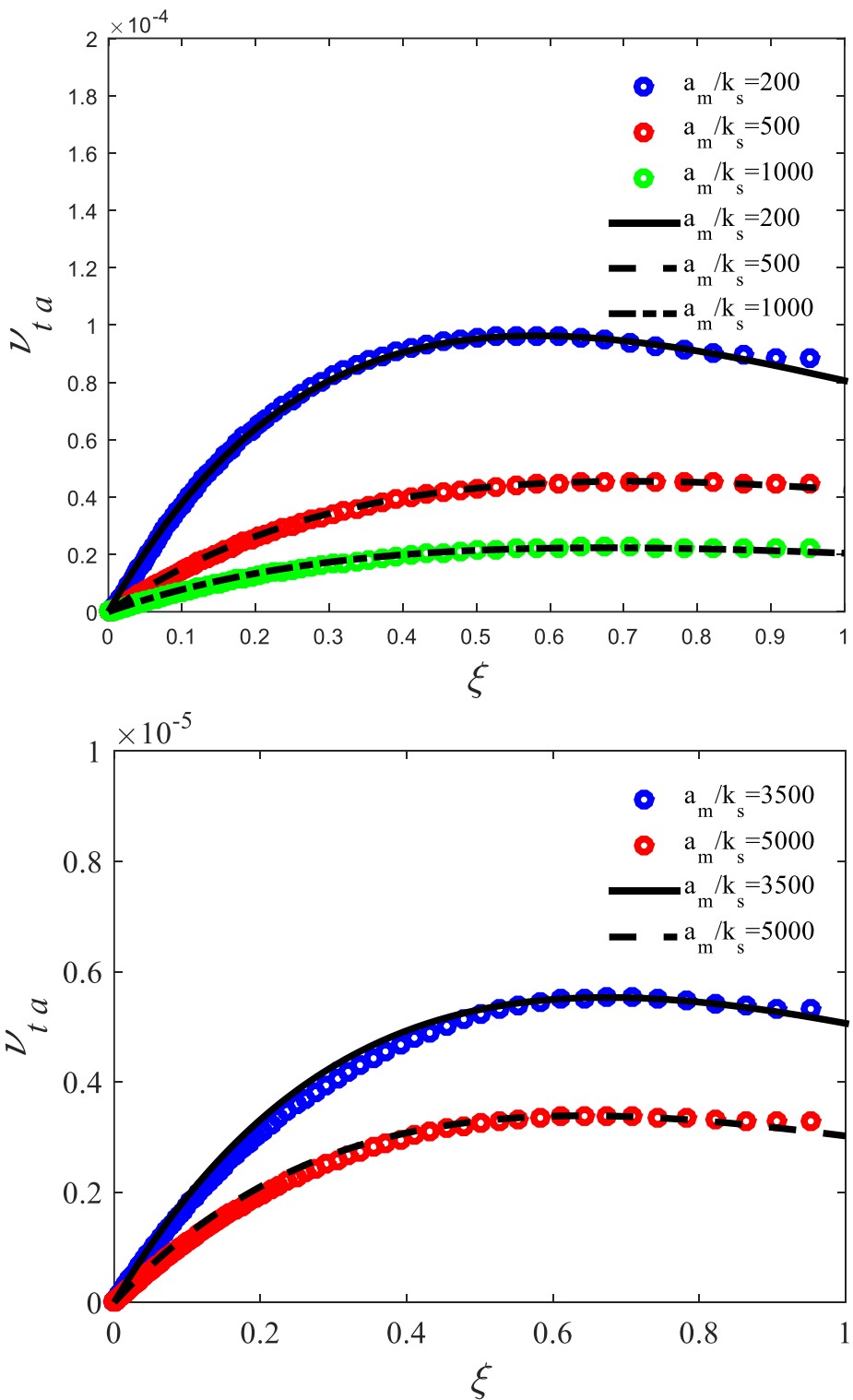

**Figure 7.** Period-averaged dimensionless eddy viscosity for a sinusoidal wave, curves: Equation (4), symbols: data from the BSL k-ω model, (**top**) $a_m/k_s$ = 200, 500, 1000 and (**bottom**) $a_m/k_s$ = 3500, 5000.

The calibration based on data from the BSL k-ω model for flow conditions up to $a_m/k_s$ = 5000 allows finding that $C_1$ becomes constant equal to 1.5 for $a_m/k_s$ = 500 (Figure 8). For $C_\alpha$, data are well described by the following equation (Figure 9)

$$C_\alpha = 0.127 \, (a_m/k_s)^{-1.061} \tag{12}$$

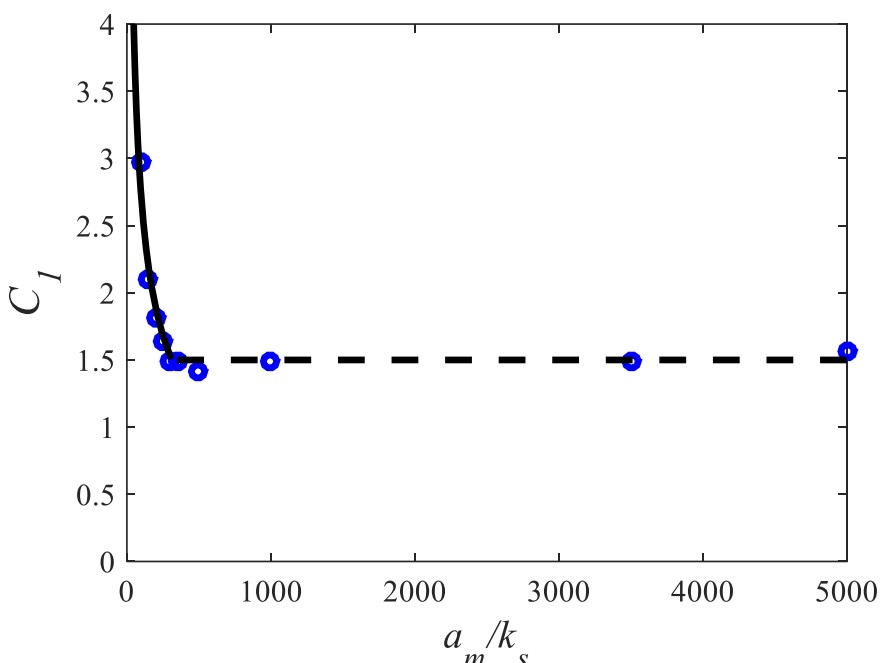

**Figure 8.** Parameter $C_1$ as a function of $a_m/k_s$, solid line: Equation (11), dashed line: constant value equal to 1.5, symbols: data from the BSL k-ω model.

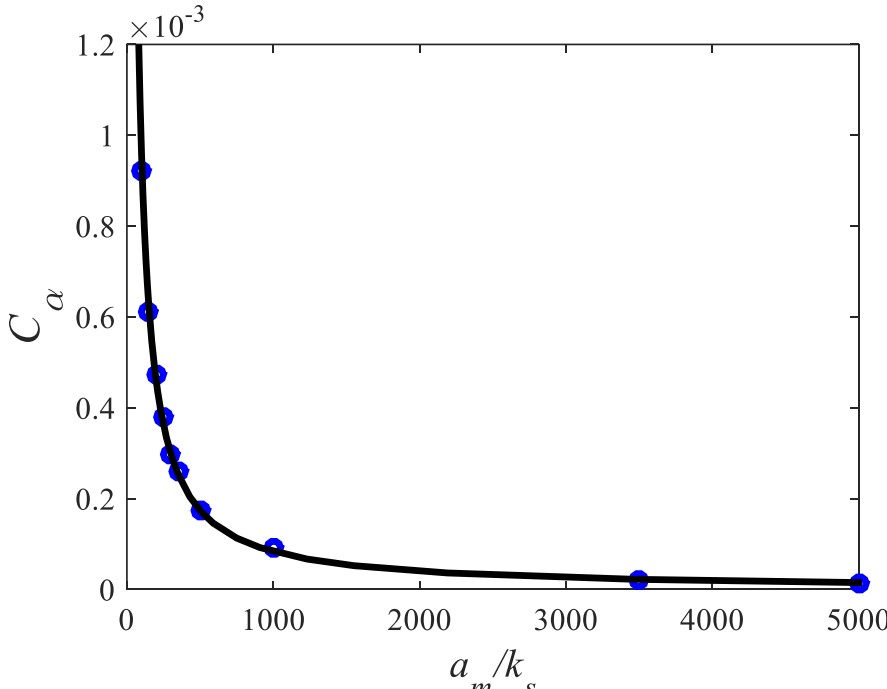

**Figure 9.** Parameter $C_\alpha$ as a function of $a_m/k_s$, curve: Equation (12), symbols: data from the BSL k-ω model.

Figure 10 shows that for high values of $k_s/a_m$, a better calibration is given by the following linear equation

$$C_\alpha = 0.0928 \, (k_s/a_m) - 4 \times 10^{-6} \tag{13}$$

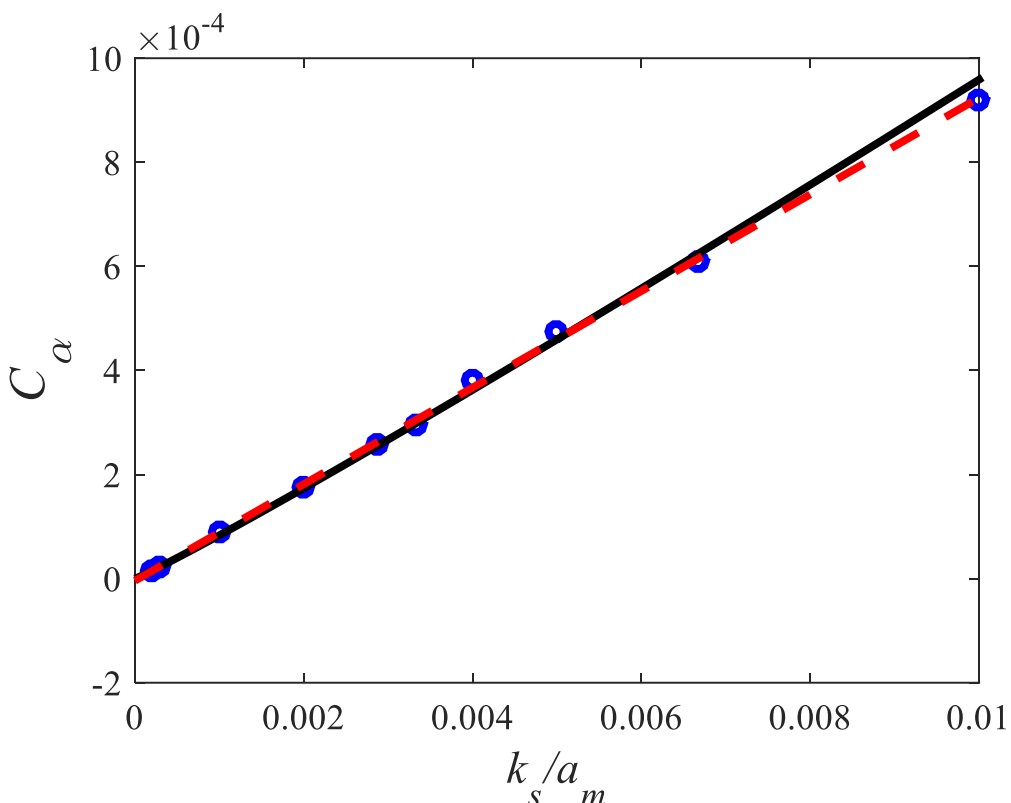

**Figure 10.** Parameter $C_\alpha$ as a function of $k_s/a_m$, black solid line: Equation (12); red dashed line: Equation (13), symbols: data from the BSL k-ω model.

Equation (13) is compared to Equation (12) which is written in the following form $C_\alpha = 0.127(k_s/a_m)^{1.061}$.

## 3. Mathematical Modeling of Suspended Sediment Concentrations

The study of the flow and transport of suspended sediments in the WBBL over sand ripples is complex and needs two-dimensional or three-dimensional space models. However, the use of these models is often too time consuming. For practical coastal engineering applications, it is preferable to use one-dimensional-vertical (1DV) models, where the parameters depend on one space variable, namely, the vertical distance from the bottom $y$ (Equations (4)–(8)), with adequate methods to parameterize the involved phenomena. In the previous section, we introduced a shape of the 1DV profile of the vertical distribution of turbulent eddy viscosity within the WBBL. However, in vortex ripples in the coastal area, in addition to the diffusive process, there is another coherent phenomenon related to vortex formation and shedding at flow reversal above ripples. Instead of the classical 1DV advection–diffusion equation (ADE), a combined 1DV convection–diffusion formulation is used with an additional term related to the convective mechanism.

### 3.1. Classical Advection–Diffusion Equation Based on the Gradient Diffusion Model

In equilibrium conditions, the concentration of suspended sediment results from the balance between an upward mixing flux $q_m$ and a downward settling flux $q_s = c(y)\,\omega_s$ as $q_m - c(y)\,\omega_s = 0$, where $\omega_s$ is the particle settling velocity and $y$ the vertical distance from the bed. The gradient diffusion model assumes that the mixing flux is proportional to the concentration gradient $q_m = -\varepsilon_s(dc/dy)$, where $\varepsilon_s$ is the sediment diffusivity and allows writing the classical 1-DV advection–diffusion equation (ADE) as

$$\varepsilon_s \frac{dc}{dy} + \omega_s c = 0 \tag{14}$$

### 3.2. Convection–Diffusion Equation with Upward Convection Term

For suspended sediments in the WBBL over sand ripples, both diffusive and convective mechanisms, related, respectively, to $\varepsilon_s dc/dy$ and vortex formation and shedding, are involved in the entrainment processes of the suspended sediments [6]. ADE (Equation (14)) is unable to predict this process of vortex formation and shedding above ripples since sediment diffusivity $\varepsilon_s$ describes only the disorganized diffusive process.

It is possible to adapt Equation (14) by adding an additional term related to the convective mechanism associated with vortex shedding at flow reversal above ripples. The steady-state ADE for the combined convection–diffusion formulation is given by [6]

$$\varepsilon_s \frac{dc}{dy} + \omega_s c + F_{conv} = 0, \tag{15}$$

The respective terms in (15) represent upward diffusion, which represents a pure disorganized "diffusive" process (given by gradient diffusion model $F_{diff} = \varepsilon_s (dc/dy)$), downward settling, and upward convection $F_{conv}$, which describes the coherent convective sediment entrainment process. Above ripples, in a ripple-averaged sense, in the lower bed/bottom part of the wave boundary layer, the convective term can dominate the upward sediment flux. However, above this bed/bottom part, the vortices lose their coherence, and the gradient diffusion becomes dominant.

The upward convection term $F_{conv}$ was given by different formulations: $F_{conv} = -\omega_s c_0 F(y)$, where $F(y)$ is a function describing the probability of a particle reaching height $y$. above the bed [6]. $F_{conv} = -\overline{v_w c_w}$, where $v_w$ and $c_w$ are periodic components, respectively, of vertical velocity and concentrations and the overbar denotes time averaging.

Both ordinary differential equations (ODEs) (Equations (14) and (15)) need the sediment diffusivity, which is the key parameter in suspended sediment concentration modeling. In Section 4, we will show that Equation (15) reverts to the classical ADE (Equation (14)) with an "apparent" sediment diffusivity.

### 3.3. Sediment Diffusivity and the Turbulent Schmidt Number

The diffusivity of sediments $\varepsilon_s$ is related to the diffusivity of momentum, i.e., the eddy viscosity $\nu_t$, by a coefficient $\beta$. The sediment diffusivity is therefore given by

$$\varepsilon_s = \beta \, \nu_t \tag{16}$$

where $\beta$ is the inverse of the turbulent Schmidt number.

Different studies were conducted toward developing equations for the turbulent Schmidt number or $\beta$-factor for both steady and oscillatory flows [42,69–75].

The method based on the finite-mixing-length model [56] allows writing the sediment diffusivity as

$$\varepsilon_s = w_m l_m \left[ 1 + \frac{l_m^2}{24} \frac{\frac{d^3 C}{dy^3}}{\frac{dC}{dy}} + \cdots \right] \tag{17}$$

With an eddy viscosity given by the product between a mixing velocity and a mixing length $\nu_t = w_m l_m$ and the assumption of an exponential decreasing concentration profile given by $c = c_b e^{-A\xi}$, Equations (8) and (9) provide an equation for $\beta(y)$ as [42]

$$\beta(y) = 1 + \frac{l_m^2}{24} A^2 \tag{18}$$

With the linear mixing length equation ($l_m = \lambda \, y$) and $\lambda = 1$ [56], (18) reverts to

$$\beta(y) = 1 + \frac{A^2}{24} y^2 \tag{19}$$

Equation (19) is similar to that proposed in [42]. Another empirical equation for $\beta(y)$ was proposed as [42]

$$\beta(y) = \beta_b f_b(y) = \beta_b e^{C_b \xi} \tag{20}$$

where $C_b$ is a coefficient. This equation presents the interest that it allows sediment diffusivity to keep the same form as Equation (4), and it will involve changing the value of the coefficient $C_1$.

The depth-averaged $\beta$-factor is obtained by integrating $\beta(y)$ over the water column as

$$\beta_{ave} = \frac{1}{h} \int_0^h \beta(y) dy = \int_0^1 \beta(\xi) d\xi, \tag{21}$$

Using Equation (19), integration of Equation (21) gives the depth-averaged $\beta$-factor as

$$\beta_{ave} = 1 + \frac{A^2}{72}, \tag{22}$$

with $A = 5.853 + 6.401 \frac{\omega}{u_*}$ [73], $\beta_{ave}$ is given by [74]

$$\beta_{ave} = 1,47 + 1.03 \left( \frac{\omega}{u_*} \right) + 0,57 \left( \frac{\omega}{u_*} \right)^2 \tag{23}$$

while with a linear function given by $A = 11 \frac{\omega}{u_*}$, $\beta_{ave}$ is given by [74]

$$\beta_{ave} = 1 + 1.68 \left( \frac{\omega}{u_*} \right)^2 \tag{24}$$

Equation (24) shows the interest that it is similar to a former empirical equation $\beta_{ave} = 1 + 2 \left( \frac{\omega}{u_*} \right)^2$ [69].

In WBBLs above ripples, cycle-mean sediment diffusivity is significantly greater than the cycle-mean eddy viscosity ($\beta > 1$). The empirical distribution of sediment diffusivity given in Figure 2 is explained nowadays as follows: The near-bed constant region is due to coherent vortex formation and shedding related to flow separation on the lee side of the steep ripple crest. In the following layer, the linearly increasing profile for sediment diffusivity is related to the random turbulent processes and gradient diffusion. Indeed, the vortices lose their coherence in this layer.

## 4. Suspended Sediments in WBBLs over Sand Ripples
### 4.1. Convection–Diffusion Model and the Classical Advection–Diffusion Equation

The convection–diffusion model for time-averaged concentrations (over the wave period) is given by Equation (15). There are two ways to write Equation (15) in the form of the ADE (Equation (14)). The first allows writing

$$\left( \frac{1}{1 + \frac{F_{conv}}{\omega_s c}} \right) \varepsilon_s \frac{dc}{dy} + \omega_s c = 0, \tag{25}$$

while the second gives

$$\left( 1 + \frac{F_{conv}}{F_{diff}} \right) \varepsilon_s \frac{dc}{dy} + \omega_s c = 0 \tag{26}$$

For both cases, the ADE is given by

$$\varepsilon_s^* \frac{dc}{dy} + \omega_s c = 0 \tag{27}$$

Equation (27) contains a modified or apparent sediment diffusivity $\varepsilon_s^*$ instead of $\varepsilon_s$, given by

$$\varepsilon_s^* = \alpha \varepsilon_s \tag{28}$$

Parameter $\alpha$ is related to the convective sediment entrainment process associated with the process of vortex shedding above ripples. Parameter $\alpha$ could be expressed by two different expressions.

In the first expression related to Equation (25), $\alpha$ is given by

$$\alpha = \frac{1}{1 + \frac{F_{conv}}{\omega_s c}} \tag{29}$$

Equation (29) shows that $\alpha$ depends on the relative importance of upward convection $F_{conv}$ related to coherent vortex shedding and downward settling of sediments $\omega_s c$. When the convective transfer is very small, for example, above low-steepness ripples, $\alpha$ becomes nearly equal to 1 ($\alpha \approx 1$), and therefore $\varepsilon_s^* \approx \varepsilon_s$. The upward convection $F_{conv}$ were given by different expressions/authors. With $F_{conv} = -\omega_s c_0 F(y)$ [6,52,53], $\alpha$ becomes equal to $\alpha = 1/(1 - (c_0/c)F(y))$, while with $F_{conv} = -\overline{v_w c_w}$ [53], $\alpha = 1/(1 - (\overline{v_w c_w}/\omega_s c))$, which seems related to the condition $(\overline{v_w c_w}/\omega_s c) < 0.2$ [45].

The second expression for $\alpha$ related to Equation (26) is given by

$$\alpha = 1 + \frac{F_{conv}}{\varepsilon_s \frac{dc}{dy}} = 1 + \frac{F_{conv}}{F_{diff}} \tag{30}$$

In Equation (30), $\alpha$ depends on the relative importance of terms $F_{conv}$ and $F_{diff} = \varepsilon_s(dc/dy)$, which are related, respectively, to coherent vortex shedding and random turbulence. When the coherent vortex shedding phenomenon is more important than random turbulence ($F_{conv} > F_{diff}$), $\alpha$ becomes greater than 1 ($\alpha > 1$). At the opposite, when random turbulence is more important than coherent vortex shedding ($F_{conv} \ll F_{diff}$), $\alpha$ becomes nearly equal to 1 ($\alpha \approx 1$), and therefore $\varepsilon_s^* \approx \varepsilon_s$.

An empirical function for $\alpha$ was proposed as [42]

$$\alpha = 1 + D \, e^{-\frac{y}{h_s}} \tag{31}$$

where $D$ and $h_s$ are two parameters. The coefficient $D$ is given by the product $D = D_R D_G$, where $D_R$ and $D_G$ are related, respectively, to the ripple shape and grain size. Above low-steepness ripples convective transfer is very small, $D_R$ becomes equal to 0, and $\alpha$ becomes equal to 1. For fine sediments, $D_G$ becomes nearly equal to 0, and therefore $\alpha \approx 1$. Time-averaged concentration profiles are obtained from the resolution of Equation (27), with an apparent sediment diffusivity $\varepsilon_s^*$ given by Equation (28), Equation (31) for $\alpha$, Equation (4) for the eddy viscosity, and Equation (20) for $\beta$. The proposed model is validated by two experimental data of oscillatory flows over sand ripples. The first are concentration profiles for fine and coarse sediments, while the second are sediment diffusivity profiles for medium sands.

*4.2. Fine and Coarse Sediments over Wave Ripples in the Same Flow (Data from [55])*

4.2.1. Experimental Conditions

In these experiments [55], suspended sediments are due to sinusoidal waves over rippled beds. Natural beach sand was used with median grain diameter $d_{50} = 0.19$ mm. The flow parameters are: wave period $T = 1.51$ s, mean wave height $H = 13$ cm, maximum near-bed flow velocity $U_m = 27.8$ cm/s, near-bed flow semi-excursion $a_m = 6.68$ cm, and mean depth of flow $h = 30$ cm. The ripples that were produced were highly uniform and regular, with a mean ripple height of $\eta_r = 1.1$ cm, a mean ripple length of $\lambda_r = 7.8$ cm, and therefore a mean ripple steepness of $\eta_r/\lambda_r = 0.14$. The equivalent roughness is given

by $k_s = 25\eta_r(\eta_r/\lambda_r)$. Measured concentrations were obtained by sieving suction samples from different elevations above the ripple crest [55].

### 4.2.2. Results

Figure 11 shows time-averaged concentration profiles, in semi-log plots, for fine and coarse sediments over sand ripples. Experimental data (symbols) show two different profiles for fine sand (o) and coarse sand (x): an upward convex concentration profile for fine sand (o) and a main upward concave profile with a near-bed upward convex profile for coarse sand (x).

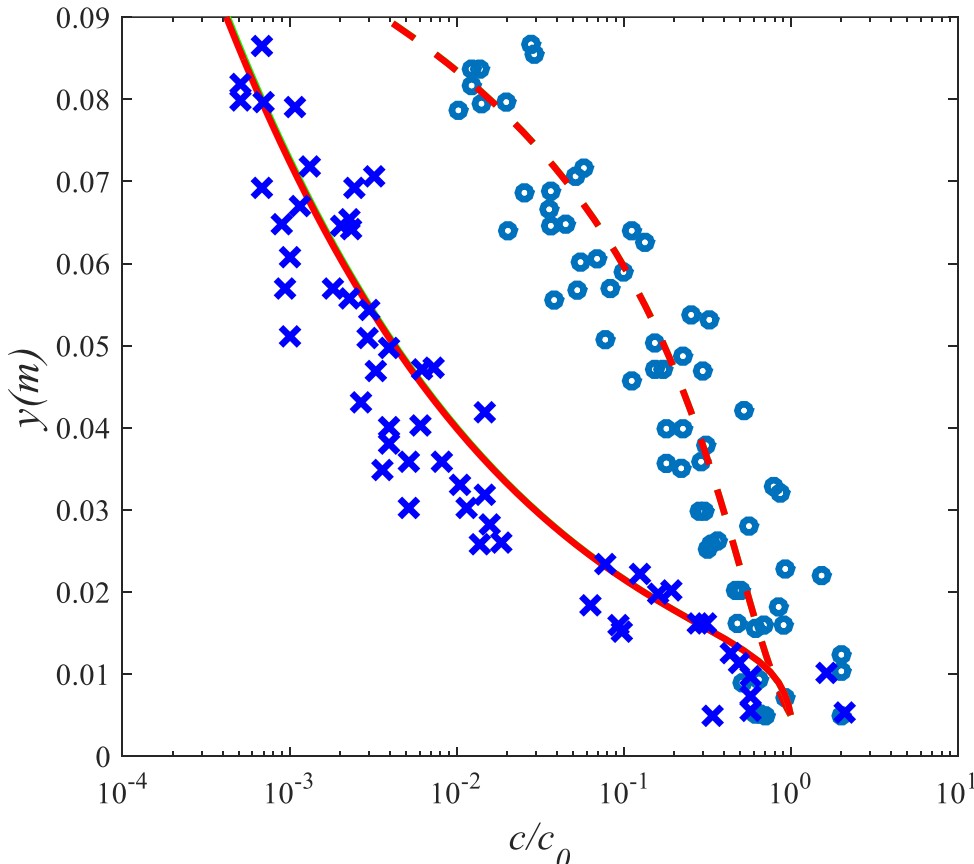

**Figure 11.** Time-averaged concentrations over wave ripples. Symbols: experimental data; (o) fine sand ($\omega_{s0}$ = 0.65 cm/s); (x) coarse sand ($\omega_{s0}$ = 6.1 cm/s), curves: solutions of Equation (27) with Equation (32).

SSC profiles (curves in Figure 11) are obtained from the numerical resolution of Equation (27) by using MATLAB. The apparent sediment diffusivity $\varepsilon_s^*$ is obtained from Equations (4), (20), (28) and (31) as

$$\varepsilon_s^* = A_s y e^{-\frac{y}{B_s}}\left(1 + D\,e^{-\frac{y}{h_s}}\right) \tag{32}$$

where $y$ is the height above the ripple crest. For fine sand, the concentration profile (dashed line) is obtained with $A_s = 0.025$ m/s, $B_s = 0.022$ m, a constant settling velocity $\omega_{s0}$= 0.65 cm/s, and $\alpha \approx 1$ (since, in Equation (31), $D_G \approx 0$). The concentration profile (dashed line) shows good agreement with experimental data (o).

For coarse sand, we take into account the effect of apparent sediment diffusivity. The solid line is computed using $\omega_{s0} = 6.1$ cm/s, $A_s = 0.017$ m/**s**, and $B_s = 0.75$ m (which correspond to $C_\alpha = 0.0538$, $C_1 = 22.38$, $\beta_b = 5.1$, and $C_b = 22$), with $D = 403$ and

$h_s = 0.002$ m. The concentration profile allows an accurate description of coarse sand (x) data (solid line in Figure 11).

This result for coarse sand was possible by using:

- The β-function (Equation (20)), which was validated by the finite-mixing-length model and allows the description of the main upward concave profile for coarse sediments
- The additional parameter $\alpha$, which allows the description of the near-bed upward convex profile. This parameter is related to the convective sediment entrainment process.

The profiles for fine and coarse sand (Figure 11) are interpreted by a relation between the second derivative of the logarithm of concentration and the derivative of the product between sediment diffusivity and $\alpha$. It is possible to write from Equation (27) [42]

$$\frac{d^2 ln(c)}{dy^2} = \frac{\omega_s}{\varepsilon_s^{*2}} \frac{d\varepsilon_s^*}{dy} \tag{33}$$

Equation (33) provides, in semi-log plots, a link between upward concavity/convexity of concentration profiles and increasing/decreasing $\varepsilon_s^*$. Increasing $\varepsilon_s^*$ allows an upward concave concentration profile, while decreasing $\varepsilon_s^*$ allows an upward convex concentration profile. Equation (33) was adapted for concentration profiles in Cartesian coordinates for steady open-channel flows [71].

*4.3. Sediment Diffusivity Profile for Medium Sediments over Ripples with Steep Slopes (Data from [53])*

We consider experimental data of near-bed sediment diffusivity over sand ripples under waves [53]. For medium sand, the ripples had relatively steep slopes. In the study [53], the form of the sediment diffusivity profiles was found to be constant with height above the bed to a height equal approximately to the equivalent roughness of the bed $k_s$. Above this level, the sediment diffusivity $\varepsilon_s$ increased linearly with height. An explanation was provided for these two layers as follows [53]:

- The constant sediment diffusivity profile is a vortex layer; the constant value of sediment diffusivity close to the bed was related to coherent vortex shedding. Steep ripples involve flow separation on the lee side of ripple crest and vortex formation.
- In the layer where the sediment diffusivity increased linearly with height, the vortices lose their coherence, and gradient diffusion becomes dominant. Random turbulent processes explain the observed linear form for $\varepsilon_s$.

In Equation (32), $y$ is the height above the ripple crest. However, concentration profiles in [55] were referenced to the undisturbed bed. Since $y$ is the height above the undisturbed bed, we write Equation (32) as

$$\frac{\varepsilon_s^*}{U_0 k_s} = A_s \frac{(\widetilde{y} - y_0)}{k_s} e^{-\frac{(\widetilde{y} - y_0)}{B_s}} \left( 1 + D\, e^{-\frac{(\widetilde{y} - y_0)}{h_s}} \right) \tag{34}$$

where $\widetilde{y}$ is the height above the undisturbed bed, and $y_0$ is the distance between the undisturbed bed level and ripple crest. Equation (34) contains two different contributions given, respectively, by parameters β (i.e., the inverse of the turbulent Schmidt number) and $\alpha$ (related to convective transfer) [54].

Figure 12 shows the mean measured normalized $\varepsilon_s^*$ (symbols) over medium sand bed and comparison with the proposed analytical profile (Equation (34)). Comparison with experimental data [53] (symbols) shows good agreement, and the shape is similar to the lower part (the two near-bed layers of sediment diffusivity, i.e., constant and linear) of the empirical profile [38] given in Figure 2.

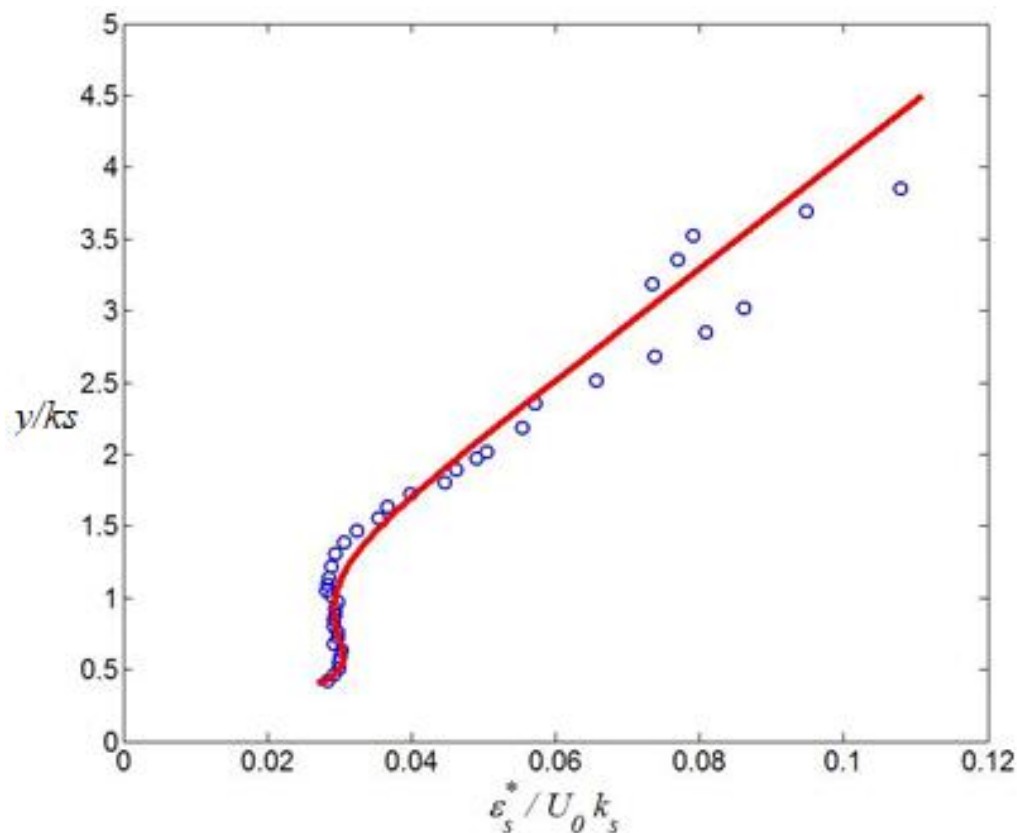

**Figure 12.** Normalized apparent sediment diffusivity over ripples with steep slopes for medium sediments, symbols: experimental data, curve: calculation from Equation (34) with $A_s = 0.026$, $y_0/k_s = 0.2$ s, $B_s/k_s = 500$, $D = 9$, and $h_s/k_s = 0.26$.

## 5. Conclusions

For the shape of the one-dimensional-vertical (1 DV) eddy viscosity profile, instead of the well-known parabolic profile, we consider the exponential-type analytical model. In this study, the exponential-type analytical model was generalized to the WBBLm and the period-averaged eddy viscosity was calibrated by a two-equation baseline (BSL) k-ω model for different flow conditions up to $a_m/k_s = 5000$.

The exponential-type analytical model was used in modeling suspended sediment concentration (SSC) profiles in oscillatory flows over sand ripples. In addition to the diffusive process, there is another coherent phenomenon related to vortex formation and shedding at flow reversal above ripples. Instead of the classical 1 DV advection–diffusion equation (ADE), a combined 1 DV convection–diffusion formulation was used with an additional term related to the convective mechanism.

Our study shows that the convection–diffusion equation reverts to the ADE with an "apparent" sediment diffusivity $\varepsilon_s^* = \alpha \, \varepsilon_s$ related to sediment diffusivity $\varepsilon_s$ by an additional parameter $\alpha$ associated with the convective sediment entrainment process over sand ripples. The additional parameter $\alpha$ was defined by two equations. In the first, $\alpha$ depends on the relative importance of upward convection related to coherent vortex shedding and downward settling of sediments. When the convective transfer is very small, above low-steepness ripples, $\alpha \approx 1$. In the second, $\alpha$ depends on the relative importance of coherent vortex shedding and random turbulence. When the coherent vortex shedding phenomenon is more important than random turbulence, $\alpha > 1$. At the opposite, when random turbulence is more important than coherent vortex shedding, $\alpha \approx 1$, and "apparent" sediment diffusivity reverts to the classical sediment diffusivity $\varepsilon_s^* \approx \varepsilon_s$.

SSC profiles are obtained from the numerical resolution of the advection–diffusion equation (ADE) with the "apparent" sediment diffusivity $\varepsilon_s^*$. The "apparent" sediment

diffusivity $\varepsilon_s^*$ is obtained by using the exponential-type analytical model, a β-function, and the function for the parameter α. Comparisons with experimental data of SSC show that the proposed method allows a good description of both concentration profiles for fine and coarse sediments. The shape of the "apparent" sediment diffusivity $\varepsilon_s^*$ profile was confirmed by experimental data and supports the two near-bed layers of sediment diffusivity (i.e., constant and linear) of the former empirical three-layer vertical distribution [38].

**Author Contributions:** R.A., conceptualization, methodology, data curation, software, validation, investigation, writing—original draft preparation; H.T., conceptualization, investigation, methodology, visualization, funding acquisition. All authors have read and agreed to the published version of the manuscript.

**Funding:** This research was partially funded by the Japan Society for the Promotion of Science (JSPS), within the "FY2021 JSPS Invitation Fellowship for Research in Japan (Short-Term)" JSPS/OF188 Fellowship ID S21074.

**Institutional Review Board Statement:** Not applicable.

**Informed Consent Statement:** Not applicable.

**Data Availability Statement:** Not applicable.

**Acknowledgments:** The authors would like to thank the three anonymous reviewers who contributed to the improvement of the manuscript. The first author is grateful for the financial support provided by the Japan Society for the Promotion of Science (JSPS). The first author would like to take this opportunity to pay homage to A.-Moumen DARCHERIF former Dean of ECAM-EPMI and Michel BÉLORGEY who both passed away in 2021.

**Conflicts of Interest:** The authors declare no conflict of interest.

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
