# Peer review of "Analytical Eddy Viscosity Model for Turbulent Wave Boundary Layers: Application to Suspended Sediment Concentrations over Wave Ripples"

_jmse, doi:10.3390/jmse11010226_

Round 1
Reviewer 1 Report (Previous Reviewer 2)
The authors have tried them best to revise this paper, this paper can be considered to be accepted.
Author Response
Thank you very for your help
Reviewer 2 Report (New Reviewer)
In this study, authors consider the exponential-type profile which was validated and calibrated by direct numerical simulation (DNS) and experimental data for turbulent channel and open-channel flows respectively. The paper provides new insights into the wave bottom boundary layer. However, there are few relevant points that should be addressed before considering publication of the manuscript.
1. Which software did you use in the paper, Fluent?
2. Model is not clear for readers. You should draw a figure to show you model in 2.2.1. Analytical eddy viscosity model.
3. More figures should be showed in 4. Suspended sediments in oscillatory flows over sand ripples. We do not see sand ripples and its changes. It must be revised.
4. Please introduce one-dimensional-vertical 318 (1DV) models in 3. Mathematical modeling of suspended sediment concentrations. It is not clear for readers.
Author Response
Dear Reviewer 2
We were pleased to know, by a letter from the Editor of Journal of Marine Science and Engineering MDPI, that our manuscript was rated as potentially acceptable for publication in Journal of Marine Science and Engineering MDPI, subject to adequate revision and response to your comments.
Please, find enclosed the revised manuscript with corrections based on your comments (in blue). Please find the response to your comments and suggestions
-----------------------------
Comments and Suggestions for Authors
In this study, authors consider the exponential-type profile which was validated and calibrated by direct numerical simulation (DNS) and experimental data for turbulent channel and open-channel flows respectively. The paper provides new insights into the wave bottom boundary layer. However, there are few relevant points that should be addressed before considering publication of the manuscript.
- Which software did you use in the paper, Fluent?
Thank you for this comment. The software used is MATLAB. We indicated it in the manuscript.
- Model is not clear for readers. You should draw a figure to show you model in 2.2.1. Analytical eddy viscosity model.
Thank you for this comment. Equations (5) and (5b) are similar to equations (2) and (3) which are shown/plotted in figure (4).
- More figures should be showed in 4. Suspended sediments in oscillatory flows over sand ripples. We do not see sand ripples and its changes. It must be revised.
Thank you for this comment. In one-dimensional-vertical (1DV) models, the parameters depend on one space variable namely the vertical distance from the bottom y (equations 5 to 8). The 1DV model provides suspended sediment concentration (SSC) (Figure 11) and apparent sediment diffusivity (Figure 12) profiles above ripple crest.
Unfortunately, 1DV model doesn’t allow seeing the three-dimensional or two-dimensional sand ripples. However, this is the topic of our current research.
- Please introduce one-dimensional-vertical 318 (1DV) models in 3. Mathematical modeling of suspended sediment concentrations. It is not clear for readers.
Thank you for this comment. This was explained in the revised manuscript.
--------------------------
As you notice, your comments were very useful and identified areas of manuscript that needed clarification. We would like to take this opportunity to express you our sincere thanks.
We hope that the revised manuscript is accepted for publication in Journal of Marine Science and Engineering MDPI.
Sincerely Yours,
The authors
Reviewer 3 Report (New Reviewer)
In general, the article is interesting, but it needs improvement in the text/description. The Authors should reduce/modify their manuscript, since they have equations and descriptions already published in previous articles. Be more concise and state clearly the innovative part of this particular work in comparison with the previous ones.
In addition, the list below addresses major issues that need consideration, and some minor points that could help with the text improvement.
Major Issues
1. L48-51. I think these lines must be deleted as they do not follow the logical structure of the paragraph.
2. L93. Which study shows…? Before this sentence you mentioned three studies [36, 43, 11].
3. L194, ξ is not defined. All the used variables must be defined and must be in italics.
4. L197, Eq. (5) and (6) is mentioned here. I think this is a typo.
5. Figure 3 is not even mentioned in the text…
6. Chapter 2.1 should be deleted. Chapter 2.2 stands on his own, since this is established in previous works [34-36, 42, 56-57].
7. Chapter 3.3. Just state the equations that you used and not write all the previous ones.
8. Equations 25 and 26 can be deleted.
9. L437, in addition of using a technical report, McFetridge, W.F.; Nielsen, P. Sediment suspension by non-breaking waves over rippled beds, Technical Report 684 No. UFL/COEL-85/005, Coast Ocean Eng Dept, University of Florida, 1985., the Authors could in addition use the concentration over ripples by https://doi.org/10.1029/2006JF000614
10. Where is equation 33 used and why is it presented?
11. L486-487, delete the sentence.
12. L492-498, why are these lines bullets?
13. L499-501, delete, or rephrase and move this sentence.
14. Delete L518-522. These are the conclusions and not an introduction or abstract.
15. Delete L524-526. This was done by previous studies.
16. L547, the empirical function for α (eq. 31) was already proposed by [42]. You cannot say that you proposed it in this article.
Minor Issues
17. L17-18. State that the exponential-type profile was validated in previous articles of yours [60-61].
18. L32-34. Is this the first equation again (L28-30)? I think you can rephrase these last sentences to be more clear.
19. L51, 61, 94, 180, oscillatory boundary layer = WBBL. Check the whole manuscript where you can use WBBL, since you defined it.
20. L61, a recent reference that you can here is https://doi.org/10.1029/2019JF005451
21. L82, k-ε etc. the variables must be in italics (check the whole manuscript).
22. L103, a recent reference that you can here is https://doi.org/10.1016/j.coastaleng.2022.104198
23. L123, a constant equal to about four. Rewrite as: equal to four.
24. L168, the bullet must start with a verb.
25. The x-axis should be νt(ξ) in Figures.
26. L219, oscillatory.
27. L223, 464, Full stop instead of comma.
28. L241, 454, 466, 551, due to instead of thanks.
29. The word “Figure” must be with capital F or not, wherever it is used. Be consistent.
30. Eq. (13) is correct?
31. L316, since 2-D and 3-D is never used again. You can omit defining them.
32. L320, define 1DV once.
33. L336 is the same with L323.
34. L359 is repeated by L342.
35. L388, follows.
36. L423, paragraph?
37. L428, void line.
38. Figure 12 must be below Eq. 34.
39. L537, define ADE once.
Author Response
Dear Reviewer 3
We were pleased to know, by a letter from the Editor of Journal of Marine Science and Engineering MDPI, that our manuscript was rated as potentially acceptable for publication in Journal of Marine Science and Engineering MDPI, subject to adequate revision and response to your comments.
Please, find enclosed the revised manuscript with corrections based on your comments (in blue).
Please find the response to your comments and suggestions
-----------------------------
Major Issues
- L48-51. I think these lines must be deleted as they do not follow the logical structure of the paragraph.
Thank you for this comment. They were deleted.
- L93. Which study shows…? Before this sentence you mentioned three studies [36, 43, 11].
Thank you for this comment. It was corrected: “these studies show” instead of “this study shows”
- L194, ξ is not defined. All the used variables must be defined and must be in italics.
Thank you for this comment. Sorry this was a mistake, it is corrected. Now all variables are in italic.
- L197, Eq. (5) and (6) is mentioned here. I think this is a typo.
Thank you for this comment. Yes, it was corrected.
- Figure 3 is not even mentioned in the text…
Thank you for this comment. Figure (3) was improved/changed and it is now mentioned in line 190.
- Chapter 2.1 should be deleted. Chapter 2.2 stands on his own, since this is established in previous works [34-36, 42, 56-57].
Thank you for this comment. In order to respond to all reviewers, chapter 2.1 was not completely deleted but it was shortened. Figure (4) bottom was removed.
- Chapter 3.3. Just state the equations that you used and not write all the previous ones.
Thank you for this comment.
- Equations 25 and 26 can be deleted.
Thank you for this comment.
- L437, in addition of using a technical report, McFetridge, W.F.; Nielsen, P. Sediment suspension by non-breaking waves over rippled beds, Technical Report 684 No. UFL/COEL-85/005, Coast Ocean Eng Dept, University of Florida, 1985., the Authors could in addition use the concentration over ripples by https://doi.org/10.1029/2006JF000614
Thank you for this comment. The article van der Werf et al. (2007) is very interesting. Unfortunately, using these data will need much time which is not possible taking into account the time allowed for resubmitting the revised manuscript. However, we will use these new data for our future work.
We added this reference 15
- van der Werf, J. J.; Doucette, J. S.: O'Donoghue, T.; Ribberink, J. S. Detailed measurements of velocities and suspended sand concentrations over full-scale ripples in regular oscillatory flow. Geophys. Res. Earth Surf. 2007, 112(F2) https://doi.org/10.1029/2006JF000614
- Where is equation 33 used and why is it presented?
Thank you for this comment. equation 33 is not used, but it allows to explain/understand the link between apparent diffusivity and concentration profiles shapes.
- L486-487, delete the sentence.
Thank you for this comment. The sentence was deleted.
- L492-498, why are these lines bullets?
Thank you for this comment. This part has been rewritten.
- L499-501, delete, or rephrase and move this sentence.
Thank you for this comment. The sentence was rephrased and moved.
- Delete L518-522. These are the conclusions and not an introduction or abstract.
Thank you for this comment. The sentence was deleted.
- Delete L524-526. This was done by previous studies.
Thank you for this comment. The sentence was deleted.
- L547, the empirical function for α (eq. 31) was already proposed by [42]. You cannot say that you proposed it in this article.
Thank you for this comment. This was deleted.
Minor Issues
- L17-18. State that the exponential-type profile was validated in previous articles of yours [60-61].
Thank you for this comment. We added this in the abstract.
- L32-34. Is this the first equation again (L28-30)? I think you can rephrase these last sentences to be more clear.
We removed the sentence “when coherent vortex shedding phenomenon is more important than random turbulence, . At the opposite, »
- L51, 61, 94, 180, oscillatory boundary layer = WBBL. Check the whole manuscript where you can use WBBL, since you defined it.
Thank you for this comment. We checked and replaced oscillatory boundary layer by WBBL.
- L61, a recent reference that you can here is https://doi.org/10.1029/2019JF005451
Thank you for this comment. We added this reference. Reference 22
- Chalmoukis, I.A.; Dimas, A.A.; Grigoriadis, D.G.E. Large-Eddy Simulation of Turbulent Oscillatory Flow Over Three-Dimensional Transient Vortex Ripple Geometries in Quasi-Equilibrium, Geophys. Res. Earth Surf. 2020, 125(8), https://doi.org/10.1029/2019JF005451
- L82, k-ε etc. the variables must be in italics (check the whole manuscript).
Thank you for this comment. All variables are now in italic.
- L103, a recent reference that you can here is https://doi.org/10.1016/j.coastaleng.2022.104198
Thank you for this comment. We added this reference. Reference 46
- Leftheriotis, G.A.; Dimas, A.A. Morphodynamics of vortex ripple creation under constant and changing oscillatory flow conditions, Eng. 2022, 177, 104198, https://doi.org/10.1016/j.coastaleng.2022.104198
- L123, a constant equal to about four. Rewrite as: equal to four.
Thank you for this comment. This is done.
- L168, the bullet must start with a verb.
Thank you for this comment. This was corrected.
- The x-axis should be νt(ξ) in Figures.
- L219, oscillatory.
Thank you for this comment. This was corrected.
- L223, 464, Full stop instead of comma.
Thank you for this comment. This was corrected.
- L241, 454, 466, 551, due to instead of thanks.
Thank you for this comment. This was corrected.
- The word “Figure” must be with capital F or not, wherever it is used. Be consistent.
Thank you for this comment. This was corrected wherever it is used.
- Eq. (13) is correct?
Thank you for this comment. Yes it is plotted in Figure 10.
- L316, since 2-D and 3-D is never used again. You can omit defining them.
Thank you for this comment. 2-D and 3-D were removed.
- L320, define 1DV once.
Thank you for this comment. This was corrected.
- L336 is the same with L323.
Thank you for this comment. This was corrected.
- L359 is repeated by L342.
Thank you for this comment. The sentence in L359 was removed.
- L388, follows.
Thank you for this comment. This was corrected.
- L423, paragraph?
- L428, void line.
Thank you for this comment. This was corrected.
- Figure 12 must be below Eq. 34.
Thank you for this comment. This was corrected.
- L537, define ADE once.
Thank you for this comment. This was corrected.
--------------------------
As you notice, your comments were very useful and identified areas of manuscript that needed clarification. We would like to take this opportunity to express you our sincere thanks.
We hope that the revised manuscript is accepted for publication in Journal of Marine Science and Engineering MDPI.
Sincerely Yours,
The authors
Round 2
Reviewer 2 Report (New Reviewer)
It is suitable to publish.
Author Response
Thank you very for your help
Reviewer 3 Report (New Reviewer)
The article has improved. However, I would like to raise the Authors attention to the following comments.
1. L28-33. Be sure that the sentences are correct, because now I read α ~ 1 at both cases (first and second). Maybe you missed an inequality?
2. L186, 196, 443, 489, Delete the space before “where”. This is not a new paragraph but just the variables definition. Check all the equations.
3. L196. I still think that the definition of ksi (ξ) is missing!
4. Eq. 13. I am still wondering if the -4x10^-6 is not the power of (ks/am), but you answered that this is not a typo…
5. L412, “shows an interest”. Rephrase this part.
Author Response
Dear Reviewer 3
We were pleased to know, by a letter from the Editor of Journal of Marine Science and Engineering MDPI, that our manuscript was rated as potentially acceptable for publication in Journal of Marine Science and Engineering MDPI, subject to adequate revision and response to your comments.
Please, find enclosed the revised manuscript with corrections based on your comments (using the “Track Changes” function). Please find the response to your comments and suggestions
-----------------------------
Comments and Suggestions for Authors
The article has improved. However, I would like to raise the Authors attention to the following comments.
- L28-33. Be sure that the sentences are correct, because now I read α ~ 1 at both cases (first and second). Maybe you missed an inequality?
Thank you for this comment. Yes the sentences are correct.
- L186, 196, 443, 489, Delete the space before “where”. This is not a new paragraph but just the variables definition. Check all the equations.
Thank you for this comment. This is done.
- L196. I still think that the definition of ksi (ξ) is missing!
Thank you for this comment. The definition of ksi (ξ) is given in L226.
- Eq. 13. I am still wondering if the -4x10^-6 is not the power of (ks/am), but you answered that this is not a typo…
Thank you for this comment. No this is not a typo, I checked the MATLAB code.
- L412, “shows an interest”. Rephrase this part.
Thank you for this comment. This is done.
--------------------------
As you notice, your comments were very useful and identified areas of manuscript that needed clarification. We would like to take this opportunity to express you our sincere thanks.
We hope that the revised manuscript is accepted for publication in Journal of Marine Science and Engineering MDPI.
Sincerely Yours,
The authors
This manuscript is a resubmission of an earlier submission. The following is a list of the peer review reports and author responses from that submission.
Round 1
Reviewer 1 Report
My major concern with this manuscript is the correctness of the analysis.
Indeed the Authors propose a numerical model to evaluate the time and space averaged sediment trasport over vortex ripples in the coastal area.
In the first part it is discussed the choice of the eddy viscosity, showing validations of the proposed model with experimental results in channel flow, which is is very different configuration with respect to that of vortex ripples in the coastal area. Indeed, it appears that many experimental data used for the validation refer to plane bed case (see figure 4b). In the sinuisodal oscillatory flow used by the authors (see eq. 4) the value of the eddy viscosity should vanish outside the bottom boundary layer, hence the profile shown in figure 4a, that extend up to the water surface is not realistic.
Incidentally the reference [38] in the caption of figure 2 is wrong, since [38] only deals with bedload, while in the present context suspended load is considered.
Reviewer 2 Report
In this paper, equation (5) is fitted through equation (6-8) calculation, and the fitting results are applied to SSC. However, only the results of equation (6-8) are given, without describing their physical significance and the detailed algorithm for solution. A simple mathematical fitting is carried out. If the author can make improvements in the following, this paper can be considered for publication:
1) This paper uses equations (6-8) to fit the turbulent viscosity to equation (5). For the calculation of equations (6-8), please give the calculation domain under a sinusoidal wave and clarify. Do you calculate in two-dimensional space or three-dimensional space? What is the initial conditions? What numerical scheme is used for calculation?
2) It is very important to write an article that others can repeat your results with the information given. Please give the numerical algorithm used for the equation, otherwise how can others judge whether your results are right or not.
3) Some words used are incorrect, for example “but for”(line95) , and there are many obviously wrong in the paper, such as listed below. Please check again.
For example:
line109, two "based on"
line143 , “kwon”
line207,”Figure(1.a)”
line211,”Figure(1.b)”
line386,”since.”
4) The conclusions (from line464) of this paper is not well written. The main contributions of the conclusions part about are unclea. It should be rewritten.